# Oxidative Effects of Potassium Dichromate on Biochemical, Hematological Characteristics, and Hormonal Levels in Rabbit Doe (*Oryctolagus cuniculus*)

**DOI:** 10.3390/vetsci6010030

**Published:** 2019-03-18

**Authors:** Chongsi Margaret Mary Momo, Ngoula Ferdinand, Ngouateu Kenfack Omer Bebe, Makona Ndekeng Alexane Marquise, Kenfack Augustave, Vemo Bertin Narcisse, Tchoffo Herve, Tchoumboue Joseph

**Affiliations:** 1Department of Animal Production, Faculty of Agronomy and Agricultural Sciences, University of Dschang, P.O. Box 188, Dschang, Cameroon; margaretchongsi@yahoo.fr (C.M.M.M.); alexanemarquise@gmail.com (M.N.A.M.); augustavekenfack@yahoo.fr (K.A.); vemobertin@yahoo.fr (V.B.N.); tchoffo.herv@yahoo.fr (T.H.); j.tchoumboue@yahoo.fr (T.J.); 2Department of Animal Biology and Physiology, Faculty of Sciences, University of Yaounde I, P.O. Box 812, Yaoundé, Cameroon; ob_ngouateu@yahoo.fr

**Keywords:** rabbit doe, potassium dichromate, oxidative stress, metabolite profile, reprotoxicity

## Abstract

The present study was conducted to evaluate the toxicity induced by the increasing doses of potassium dichromate in rabbit doe. Twenty-eight adult does of 6 months of age were divided into four groups (A, B, C, and D; n = 7), with comparable average body weight (bw). Group A rabbits received only distilled water daily and served as a control, while groups B, C, and D received, respectively, 10 mg/kg bw, 20 mg/ kg bw, and 40 mg/kg bw of potassium dichromate via gavage for 28 days, after which animals were anesthetized with ether vapor and sacrificed. Blood samples were obtained via cardiac puncture and collected without anticoagulant for biochemical dosages and with anticoagulant (EDTA) for complete blood count. Follicle stimulating hormone (FSH), luteinizing hormone (LH), and estradiol (E2) were dosed in serum and in homogenates of ovary with the help of AccuDiag^TM^ ELISA kits from OMEGA DIAGNOSTICS LTD (Scotland, England) while respecting the immuno-enzymatic method. Activities of superoxide dismutase (SOD), glutathione (GSH), catalase (CAT), and concentration of malondialdehyde (MDA) in liver, kidney, ovary and uterus were measured. Hematology revealed a significant (*p* < 0.05) decrease in mean values of hemoglobin and platelets while white blood cells and lymphocytes showed a significant (*p* < 0.05) increase in exposed groups. No significant (*p* > 0.05) difference was registered in monocytes, red blood cells, hematocrits, and plaquetocrits values with respect to the control. No matter the organ considered, no significant (*p* > 0.05) change was recorded in weight and volume. Nephrotoxicity analysis registered a significant (*p* < 0.05) increase in urea and creatinine, unlike renal tissue protein, which decreased significantly (*p* < 0.05). However, hepatotoxicity registered no significant (*p* > 0.05) variation in aspartate aminotransferase but total protein, alanine aminotransferase, and total cholesterol increased significantly (*p* < 0.05), while hepatic tissue protein revealed a significant (*p* < 0.05) decrease. Analysis on reproductive parameters showed a significant (*p* < 0.05) decrease in ovarian and uterine tissue proteins, as well as in follicle stimulating hormone, luteinizing hormone, and estradiol. Oxidative stress markers recorded no significant (*p* > 0.05) difference in glutathione reductase except in ovary where a significant (*p* < 0.05) decrease was seen when compared with the control, while catalase revealed a significant (*p* < 0.05) decrease, except in liver where there was no significant (*p* > 0.05) change. Superoxide dismutase and malondialdehyde recorded a significant (*p* < 0.05) decrease and increase respectively, with respect to the control. Results obtained from this study showed that the reduction process of chromium in tissues may cause the generation of reactive oxygen species, which are involved in hematoxic, nephrotoxic, hepatotoxic, and reproductive toxicity effects.

## 1. Introduction

Many studies [1,2,3,4] have reported toxic and carcinogenic effects induced when humans and animals are exposed to certain heavy metals. Detailed studies have shown that metals like iron, copper, cadmium, chromium, mercury, nickel, and vanadium possess the ability to produce reactive oxygen species (ROS), resulting in lipid peroxidation, depletion of proteins, and many other effects [5]. Potassium dichromate, which is a form of hexavalent chromium, has been demonstrated to induce toxicity associated with oxidative stress in humans and animals [6,7]. Chromium (Cr) is a naturally occurring element found in rocks, animals, plants, soil, and in volcanic dust and gases [8,9]. It comes in several different forms including trivalent chromium and hexavalent chromium or Cr(VI). It is widely used in various industries, including pigments for manufacturing and painting, metal plating and leather tanning. Cr(VI) ingested with food, such as vegetables or meat and water, is reduced to Cr(III) before entering the bloodstream [10,11]. Chromium enters the body through the lungs, gastrointestinal tract, and to a lesser extent, through the skin [12,13].

It is known that oral intake, including food and water, is the major route of exposure to chromium for the general population. Regardless of the route of exposure, Cr(III) is poorly absorbed, whereas Cr(VI) is more readily absorbed [14,15]. During reduction process, Cr produces ROS [16], and generates oxidative stress (OS). Multiple studies of the developmental toxicity of Cr(VI) in experimental animal models, such as rats and mice, have demonstrated a decrease in the viability of the conceptus, both pre- and post-implantation resorptions, decrease in fetal weights and crown-rump lengths, changes in placental weights (decrease or increase), and increase in frequencies of external and skeletal anomalies [17,18,19,20,21]. 

The toxicity of potassium dichromate has not been well-studied in farm animals over the years, and therefore the scarcity of studies confirming its effects on these animals have revealed the need to carry out toxicity studies on it. Thus, the objective of the present work was to evaluate the oxidative effects of potassium dichromate on biochemical, hematological, and reproductive parameters in female rabbits.

## 2. Materials and Methods

### 2.1. Animals

Twenty-eight adult does (New Zealand breed) of 6 months old, weighing 2.8–3.0 kg, reproduced at Teaching and Research Farm of the University of Dschang, were used. The animals were maintained individually in cages of wire netting (96 cm long, 40 cm wide, and 15 cm high) and in galvanized metal, forming a battery of cages. These cages were each equipped with an eater and a drinker. The animals received water and feed *ad libitum*. The composition and chemical characteristics of this feed are summarized in Table 1. Experimental protocols used in this study were approved by the Department of Animal Science, FASA, University of Dschang - Cameroon and strictly conformed with the internationally accepted standard of ethical guidelines for laboratory animal use and care as described in the European Community guidelines; EEC Directive 86/609/EEC, of the 24 November 1986.

### 2.2. Chemicals

Potassium dichromate (K_2_Cr_2_O_7_) is a compound that is prepared from chromite (FeCr_2_O_4_) and is also called chrome in iron ore. Its atomic weight is 51.996 [22]. This chemical was obtained from Sigma Aldrich, Berlin, Germany.

### 2.3. Toxicological Assessment

Twenty-eight animals were divided into four groups of seven females, with comparable average body weight. Group A (the control) received 1 mL of distilled water, while groups B, C, and D received doses of 10, 20, and 40 mg/kg body weight (bw) K_2_Cr_2_O_7_, respectively, via gavage for 28 days. Rabbits were observed during the first 24 h for the beginning of any immediate toxic signs and daily for 28 days. At the end of the experiment, animals were anesthetized with ether vapor. 

### 2.4. Hematological and Biochemical Analysis

Blood samples were obtained by cardiac puncture and collected without anticoagulant for biochemical dosages and with anticoagulant (EDTA) for complete blood count. The biochemical parameters analyzed from serum were total cholesterol (TC), aspartate aminotransferase (AST), alanine aminotransferase (ALT), urea (Ur), creatinine (Cr), Albumin (Al), and total protein (TP), performed with appropriate commercial Chronolab kits (Barcelona, Spain). The spectro-photometric method was used according to kits instructions. The hematological parameters analyzed were white blood cells (WBC), lymphocytes (LY), monocytes (MO), granulocytes (GR), red blood cells (RBC), platelets (PLT), plaquetocrit (PCT), hematocrit (HCT), and hemoglobin (Hb), performed using a veterinary hematology analyzer Genius KT 6180 (Shenzhen Genius Electronics Co., Ltd., Hong Kong, China). Follicle stimulating hormone (FSH), luteinizing hormone (LH), and estradiol (E2) were dosed in serum and in homogenates of ovary. This dosing was realized with the help of AccuDiag^TM^ ELISA kits from OMEGA Diagnostics Ltd (Alva, United Kingdom) respecting the immuno-enzymatic method.

### 2.5. Weight and Volume of Organs

After animal sacrifice the right ovary, kidney, liver, and uterus were removed and weighed. 

### 2.6. Oxidative Stress Markers

Activities of superoxide dismutase (SOD), glutathione (GSH), catalase (CAT), and concentration of malondialdehyde (MDA) in liver, kidney, ovary, and uterus were measured.

#### 2.6.1. Estimation of Catalase Activity

Catalase activity was estimated according to Aebi [23], depending on the ability of H_2_O_2_ to decompose via the action of CAT to produce H_2_O and O_2_. The decrease in the absorbance in the UV region per time is corresponded to CAT activity. A volume of 2.0 mL of substrate (10 pmol/mL of H_2_O_2_ in 50 mmol/L sodium-potassium phosphate buffer, pH 7.0) was incubated with 100-µL serum. The decomposition of H_2_O_2_ was followed directly for 2 mins by the decrease in absorbance at 240 nm.

#### 2.6.2. Estimation of Glutathione 

Glutathione level was measured according to the method of Moron et al., [24] based on the reaction of GSH with 5,5-dithiobis-2-nitro-benzoic acid (DTNB)(Sigma Aldrich, Berlin, Germany) at pH 8.0 was added to the tubes and the intense yellow color formed was red at 412 nm in a spectrophotometer after 10 min. A standard curve of GSH was prepared using concentrations ranging from 2 to 10 nmol of GSH in 5% trichloroacetic acid (Labtech, Windsor, Australia).

After centrifugation, the absorbance of yellow color was measured and the results were calculated from the glutathione standard curve.

#### 2.6.3. Estimation of Lipid Peroxidation 

Lipid peroxidation was estimated by reaction of thiobarbituric acid (TBA) (from Qulaikems, New Delhi, India) with malondialdehyde (MDA) according to Botsoglou et al., [25]. In the presence of an acid and heat (pH 2–3, 100 °C), MDA condensed with two molecules of TBA to produce a pink color complex that absorbs at 532 nm. A total of 105 μL of orthophosphoric acid at 1% and 500 μL of the precipitation mixture (1% TBA in a 1% acetic acid solution) was added to 100 μL homogenate. The mixture of each tube was homogenized and placed in a boiling water for 15 min. The tubes were cooled in an ice-bath and the mixture was centrifuged at 3500 rpm for 10 min. Absorbance was read at 532 nm against the control.

#### 2.6.4. Estimation of Superoxide Dismutase

Adrenaline is stable enough when pH is acidic. When pH increases, the rate of auto-oxidation of adrenaline increases. The dosage of SOD is thus based on the capacity of SOD to inhibit or slow down auto-oxidation of adrenaline to adreno-chromium in a milieu of a base. The method proposed by Misra and Fridovich [26] was used in this study. Microtubes of serum were introduced into the spectrometer, as well as 1660 µL of carbonate buffer solution (pH = 10) and 200 µL of adrenaline (0.3 mM). The absorbance of adreno-chromium formed was read at 480 nm 30 and 90 s after the initiation of the reaction. 

### 2.7. Statistical Analysis

Data were submitted to analysis of variance (ANOVA) one factor to test the effect of K_2_Cr_2_O_7_ on the studied parameters. Duncan’s test was used to separate means when there was a significant difference. The results were expressed in the form of mean ± standard deviation. The limit of significance was set at 5% and the software IBM SPSS Statistics 20.0, (Armonk, NY, USA) was used for analysis.

## 3. Results

### 3.1. Effects of Potassium Dichromate on Indicators of Toxicity in Rabbit Does

#### 3.1.1. Effects of Potassium Dichromate on Live Body Weight in Rabbit Does

It appears from Figure 1 that body weight increased significantly (*p* < 0.05) throughout the experimental period in does that received distilled water only. Meanwhile, whatever the dose of potassium dichromate, a significant decrease was registered in these groups in a dose-dependent manner.

#### 3.1.2. Kidney Weight and Volume and Biochemical Markers of Nephrotoxicity

The effects of potassium dichromate on kidney weight and volume, and biochemical markers of nephrotoxicity in rabbit does are represented in Table 2. The weight and volume of kidney showed no significant (*p* > 0.05) difference as compared to the control group, unlike creatinine and urea, which showed a significant (*p* < 0.05) increase with respect to the control group. A significant (*p* < 0.05) decrease was observed in renal tissue protein when compared to the control group.

#### 3.1.3. Liver Weight and Volume and Biochemical Markers of Hepatotoxicity

Table 3 summarizes the effects of potassium dichromate on liver weight and volume as well as the biochemical markers of hepatotoxicity in rabbit doe. It can be noticed here that whatever the dose, no significant (*p* > 0.05) difference was registered in weight and volume of the liver, AST, and total protein in comparison with the control group. On the other hand, statistical analysis revealed a significant (*p* < 0.05) increase in the values of ALT and total cholesterol with increasing doses as compared to the control group, though the difference between the low dose and the control group was not significant (*p* > 0.05). However, a significant decrease with increasing doses was registered in hepatic tissue proteins with respect to the control, though the difference between the mid and high doses was insignificant. A significant (*p* < 0.05) decrease was recorded in albumin with increasing doses with the lowest value in high dose

### 3.2. Effects of Potassium Dichromate on Reproductive Toxicity Indicators in Rabbit Does

#### 3.2.1. Ovary Weight and Tissue Proteins

The effects of potassium dichromate on ovary weight and tissue proteins in rabbit doe are shown in Table 4. It was noticed here that, independent of the dose, no significant (*p* > 0.05) difference was recorded in ovary weight; this is unlike the case with ovarian and uterine tissue proteins where a significant (*p* < 0.05) decrease was registered with increasing doses of potassium dichromate in comparison with the control.

#### 3.2.2. Reproductive Hormones

##### Serum Concentration of FSH

Figure 2 shows the effects of potassium dichromate on serum concentration of FSH in rabbit does. It was observed here that the serum concentration of FSH decreased significantly (*p* < 0.05) with increasing doses in rabbit does submitted to potassium dichromate when compared with the control.

##### Serum Concentration of LH

The effects of potassium dichromate on serum concentration of LH in rabbit does is summarized in Figure 3. From this figure, we noticed a significant (*p* < 0.05) decrease in serum concentration of LH with the different doses of potassium dichromate as compared to does that received distilled water.

##### Serum Concentration of Estradiol

The effects of potassium dichromate on serum concentration of estradiol are represented in Figure 4. It appeared from this figure that there was a significant (*p* < 0.05) decrease in serum concentration of estradiol in the potassium dichromate-treated does in comparison with the control group.

### 3.3. Effects of Potassium Dichromate on Hematological Parameters in Rabbit Does

Increasing oral administration of potassium dichromate produced various hematological alterations in rabbits (Table 5). WBC and lymphocytes were significantly (*p* < 0.05) increased with increasing doses of potassium dichromate with respect to the distilled water-receiving group. There was a significant (*p* < 0.05) decrease in hemoglobin in does submitted to potassium dichromate as compared to the control group. A significant (*p* < 0.05) decrease was seen in platelets with the different doses of potassium dichromate as compared to the control. Independently of doses, monocytes, RBC, hematocrit, and plaquetocrit were comparable (*p* > 0.05) among all groups.

### 3.4. Effects of Potassium Dichromate on Oxidative Stress Markers in Rabbit Does

The effects of potassium dichromate on oxidative stress markers are shown in Table 6. It appears from this table that in all the organs considered, there was no significant (*p* > 0.05) difference in GSH contents except in the ovary where there was a significant (*p* < 0.05) decrease with respect to the control group. A significant (*p* < 0.05) decrease in SOD activity was registered in all groups treated with potassium dichromate with the lowest value found in the group with the highest dose when compared to control, no matter the organ. CAT activity registered a dose-dependent significant decrease in all the organs considered when compared with the control group, except in the liver where no significant difference was noticed. Whatever the organ considered, there was a significant (*p* < 0.05) increase in MDA with respect to the control group.

## 4. Discussion

The exposure of animals to heavy metals can lead to adverse effects on their health in general and on their reproduction in particular. Kidney and liver are organs very important in the evaluation of the toxic potential of a substance [27]. They are associated with the metabolism and excretion of toxic substances [28] such as heavy metals. In the current study, live body weight decreased significantly with increasing doses of potassium dichromate throughout the experimental period. This result is in accordance with those of [29,30]. This may be due to the effects of potassium dichromate on female rabbits. The weight and volume of kidney and liver, as well as ovary weight, were comparable among treatments. This observation could signify that the dose of potassium dichromate administered did not have a pronounced effect on the anatomy of those organs. These results do not agree with Saha et al. [31], who recorded a significant reduction in the weight of the reproductive organs along with an increase in the weight of the liver and kidney; and Petrovici et al. [32], who noted an increase in ovary weight of rats exposed to potassium dichromate at a dose of 10 mg/kg bw with respect to the control.

Kidneys perform two major important functions: first, they excrete most of the end products of bodily metabolism, and second, they control the concentrations of most of the constituents of the body fluids [33]. As markers of renal function, uric acid, urea, and creatinine are routinely used for analysis. In kidneys, urea is filtered out of blood by glomeruli and is partially reabsorbed with water [34]. Creatinine is a breakdown product of creatine phosphate in muscle and is usually produced at a constant rate by the body depending on muscle mass [35]. The biochemical analyses in this study showed an increase in the concentrations of creatinine, urea, and total cholesterol levels, with a decline in proteins and albumin levels. Albumin is a specific protein and its decrease in this study can be associated with the decrease in proteins. This decrease reflects damage in hepatocytes and indicated the general and systemic toxic effect of heavy metals on rabbit does [36]. Similar results were reported by Saha et al., [31] in rats treated with potassium dichromate. An increase in cholesterol level in this study might be due to less utilization of these nutrients at the tissue level [37]. The increase in creatinine and urea could be due to the dysfunction of glomerulus, which are the structures responsible for the renal filtration. That of urea could also be explained by the increase in proteins’ catabolism, due to the high synthesis of the enzyme arginase, which intervenes in the urea production [38]. Transaminases and lactate dehydrogenase are the most sensitive biomarkers directly implicated in the extent of cellular damage and toxicity because they are localized in the cytoplasm and are released into the circulation after cellular damage [39]. 

A high level of serum of enzymes alanine aminotransferase (ALT) and aspartate aminotransferase (AST) indicate liver damage, such as that due to viral hepatitis, as well as cardiac infraction and muscle injury. Serum ALT catalyzes the conversion of alanine to pyruvate and glutamate. Therefore, serum ALT is more specific to the liver, and is thus a better parameter for detecting liver injury [40]. Their increase in the current study suggests that potassium dichromate could have generated reactive oxygen species, thus oxidative stress leading to hepatotoxicity and nephrotoxicity. This may be due to the impairment in their synthesis or poor liver function associated to oxidative stress [41]. Similar results were observed by Abbas and Ali, Zhu et al., Mehany et al., Mohamed and Saber, Saha et al., and Krim et al. [31,42,43,44,45,46] in rats treated with potassium dichromate.

Blood acts as a pathological reflector of the status of exposed animals to toxicants and other conditions and/or agents [47]. Monocytes, granulocytes, and lymphocytes, which are associates of white blood cells, act as defenders against pathogens and tend to increase in case of infection. Thus, the increase in white blood cells in this study can be explained by the fact that potassium dichromate was recognized as a toxicant. Hemoglobin is related to the total population of red blood cells in blood. The decrease in hemoglobin suggests that potassium dichromate could have blocked the erythropoietin release, which is the humoral regulator of red blood cell production. The similar decrease in platelets can be attributed to the blockage of thrombopoietin. The present results are in agreement with the findings of Shrivastava et al., Stana et al., Ounassa, Vihol et al. [48,49,50,51] in rats and mice treated with potassium dichromate.

The hypothalamic–pituitary–gonadal (HPG) axis plays a critical role in the control of reproduction. Potassium dichromate exposure led to the decrease in follicle stimulating hormone (FSH), luteinizing hormone (LH), and estradiol. This might be due to the decrease in protein levels. In fact, FSH and LH are hormones synthesized from proteins captured from blood. Thus, the reduction in total proteins in blood could have led to a decrease in FSH and LH concentrations in potassium dichromate-treated rabbit does. The abnormal levels of sex hormones recorded in this study are in accordance with those registered in female rats exposed to potassium dichromate, by Assasa and Farahat [52] suggesting a disruption of steroidogenic function. Animals in this study are of the same age and in principle age is not therefore a factor of variation in hormonal level in this case. Considering other studies, such as that carried out by Zhang et al. [53] in New Zeeland female rabbits and Labib et al. [54], age and corpus luteum formation can be a factor of hormonal expression. 

The changes in oxidative stress biomarkers have been reported to be an indicator of a tissue’s ability to cope with oxidative stress [55]. The antioxidant enzyme catalase (CAT) acts as defence against free radicals. It is responsible for the catalytic decomposition of hydrogen peroxide to molecular oxygen and water. Glutathione (GSH) is normally present in millimolar concentrations in cells and is known to protect the cellular system against the toxic effects of lipid peroxidation. It is very important in maintaining cellular redox status [56] and its depletion is considered as a marker of oxidative stress [57]. The decreased superoxide dismutase (SOD) activity may lead to massive production of the superoxide anion. The production of such anions overrides enzymatic activity and leads to a fall in its concentration in renal tissue. It was indicated by Srinivasan et al. and Pedraza-Chaverri et al. [58,59] that most of the antioxidant enzymes become inactive after potassium dichromate exposure, either due to the direct binding of heavy metals to enzyme active site or to the displacement of metal co-factors from active sites. Increased lipid peroxidation is indicated by the increase in malondialdehyde (MDA), which is an end-product of lipid peroxidation. The current results showed that treatment with potassium dichromate induced oxidative stress notified by a significant decrease in GSH content, SOD, and CAT activities, and a significant increase in MDA level, as compared to control values. The results obtained for the various oxidative stress biomarkers in the present study reflect those obtained by Mehany et al. [44], Shati [60] and Mohamed and Saber, [45]. 

## 5. Conclusions

In conclusion, the present study proved that potassium dichromate administration to does led to a significant increase in the levels of creatinine, urea, alanine aminotransferase, and total cholesterol, with a significant decrease in renal tissue proteins, hepatic tissue proteins, and albumin levels, as compared with the control group. This may be evidence for its hepatotoxic and nephrotoxic effects.

A significant decrease in follicle stimulating hormone, luteinizing hormone, and estradiol levels was equally observed in potassium dichromate-treated does with reference to the control females, indicating its dangerous effects on reproductive hormones. 

The administration of potassium dichromate brought about a significant increase in white blood cells and lymphocyctes, with a significant decrease in hemoglobin levels with respect to does that received distilled water only. This could be a confirmation of its hematotoxic activities in female rabbits.

Does treated with potassium dichromate only registered a significant decrease in glutathione, superoxide dismutase, and catalase activities, while malondialdehyde increased significantly relative to the control does. This could be attributed to its deleterious effects on enzymatic antioxidants and its lipid peroxidation activities. The observed effects of potassium dichromate in this study showed that it had dose-dependent intoxication.

## Figures and Tables

**Figure 1 vetsci-06-00030-f001:**
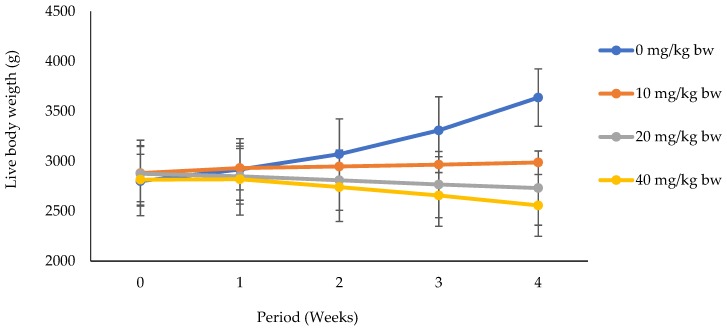
Weekly evolution of live weight in rabbit does exposed to potassium dichromate with 0, 10, 20, and 40 mg/kg bw doses of potassium dichromate.

**Figure 2 vetsci-06-00030-f002:**
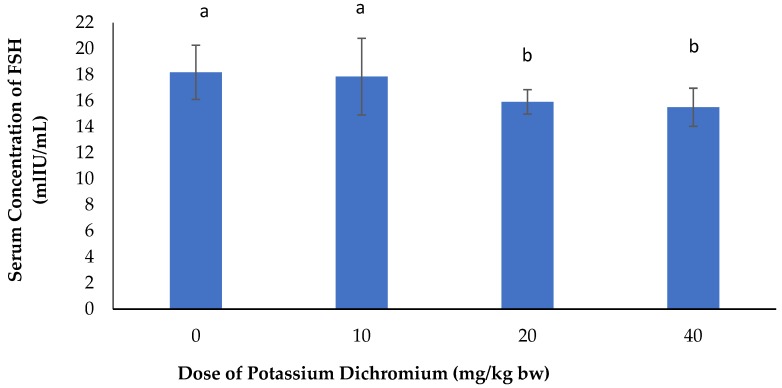
Effects of potassium dichromate on serum concentration of FSH in rabbit does. a,b: bars affected with the same letter do not differ significantly (*p* > 0.05).

**Figure 3 vetsci-06-00030-f003:**
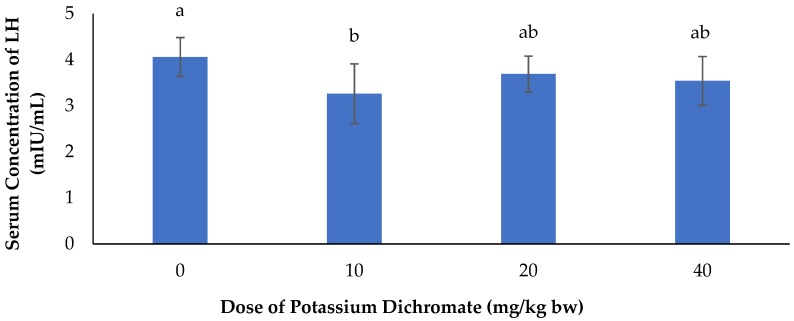
Effects of potassium dichromate on serum concentration of LH in rabbit does. a,b: bars affected with the same letter do not differ significantly (*p* > 0.05).

**Figure 4 vetsci-06-00030-f004:**
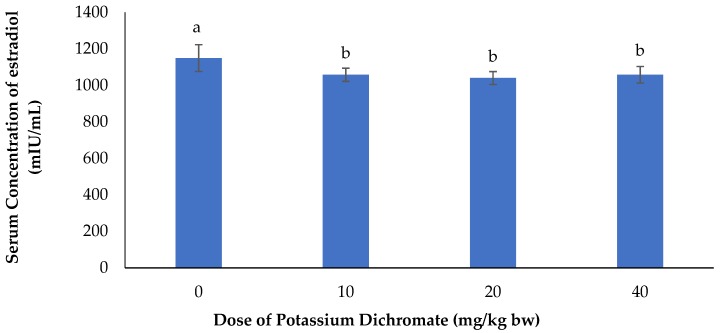
Effects of potassium dichromate on serum concentration of estradiol in rabbit does. a,b: bars affected with the same letter do not differ significantly (*p* > 0.05).

**Table 1 vetsci-06-00030-t001:** Composition and chemical characteristics of the feed.

Ingredients	Quantities (%)
Maize	27.00
Wheat bran	14.00
Kernel cake	18.00
Soy beans cake	5.00
Cotton cake	4.00
Premix10% *	5.00
Fish meal	3.00
Palm oil	2.00
Sea shells	1.50
Salt	0.50
Rice bran	20.00
Total (kg)	100.00
**Chemical Characteristics**	
Metabolizable energy (kcal/kg)	2435.23
Crude proteins (% DM)	16.47
Crude cellulose (% DM)	13.65
Calcium (% DM)	1.26
Phosphorus (% DM)	0.55
Sodium (% DM)	0.28
Lysine (% DM)	0.83
Methionine (% DM)	0.36

Premix10% *: mixture of vitamins A, B complex, D, K, and E plus Iron, Cu, Zn, Se, Mn, methionine, and lysine principally and incorporated at 5% in diet.; DM: dried matter

**Table 2 vetsci-06-00030-t002:** Effects of potassium dichromate on kidney weight and volume and biochemical markers of nephrotoxicity in rabbit does.

Toxicity Parameters	Doses of Potassium Dichromate (mg/kg bw)				*p*
	0	10	20	40	
**Weight of the Kidney (g/100 g of bw)**	0.51 ± 0.04	0.47 ± 0.02	0.48 ± 0.06	0.51 ± 0.01	0.52
**Volume of the kidney (mL)**	12.50 ± 0.87	12.33 ± 0.58	13.00 ± 1.73	12.67 ± 1.15	0.91
**Creatinine (mg/dL)**	1.72 ± 0.17 ^b^	2.25 ± 0.30 ^a^	2.01 ± 0.27 ^ab^	2.18 ± 0.17 ^a^	0.02
**Urea (mg/dL)**	24.65 ± 7.86 ^b^	37.76 ± 7.34 ^a^	43.59 ± 8.09 ^a^	38.26 ± 3.74 ^a^	0.00
**Renal Tissue Protein (g/dL)**	1.52 ± 0.26 ^a^	0.63 ± 0.16 ^b^	0.53 ± 0.14 ^b^	0.70 ± 0.13 ^b^	0.00

a,b: values with the same letter per row are not significantly different (*p* > 0.05).

**Table 3 vetsci-06-00030-t003:** Effects of potassium dichromate on liver weight and volume and biochemical markers of hepatotoxicity in rabbit does.

Toxicity Parameters	Doses of Potassium Dichromate (mg/kg bw)				*p*
	0	10	20	40	
**Liver weight (g/100 g of bw)**	2.30 ± 0.37	2.10 ± 0.27	2.08 ± 0.28	2.74 ± 0.63	0.25
**Volume of the liver (mL)**	55.67 ± 10.07	51.67 ± 7.64	56.67 ± 11.55	68.33 ± 12.58	0.32
**ALT (IU)**	40.03 ± 8.88 ^c^	46.81 ± 7.68 ^c^	62.02 ± 9.96 ^b^	74.38 ± 8.53 ^a^	0.00
**AST (IU)**	54.69 ± 10.24	69.13 ± 14.94	66.21 ± 12.39	69.34 ± 10.26	0.31
**Total cholesterol (mg/dL)**	35.87 ± 7.55 ^b^	45.65 ± 6.00 ^ab^	54.45 ± 11.48 ^a^	44.69 ± 11.49 ^ab^	0.04
**Total protein (g/dL)**	4.68 ± 0.61	4.79 ± 0.65	4.60 ± 0.54	4.11 ± 0.63	0.42
**Albumin (g/dL)**	4.34 ± 0.38 ^a^	3.94 ± 0.54 ^ab^	3.71 ± 0.59 ^ab^	3.47 ± 0.46 ^b^	0.05
**hepatic tissue protein (g/dL)**	2.58 ± 0.53 ^a^	1.67 ± 0.27 ^b^	0.96 ± 0.27 ^c^	0.74 ± 0.27 ^c^	0.00

a,b,c: values with the same letter per row are not significantly different (*p* > 0.05). AST: aspartate aminotransferase, ALT: alanine aminotransferase.

**Table 4 vetsci-06-00030-t004:** Effects of potassium dichromate on ovary weight and tissue proteins in rabbit doe.

Toxicity Parameters	Doses of Potassium Dichromate (mg/kg bw)				*p*
	0	10	20	40	
**Ovary weight (g/100 g of bw)**	0.02 ± 0.01	0.02 ± 0.00	0.02 ± 0.01	0.02 ± 0.00	0.87
**Ovarian tissue protein (g/dL)**	2.14 ± 0.30 ^a^	1.24 ± 0.27 ^b^	1.17 ± 0.33 ^b^	0.63 ± 0.13 ^c^	0.00
**Uterine tissue protein (g/dL)**	0.42 ± 0.08 ^a^	0.33 ± 0.03 ^ab^	0.27 ± 0.08 ^bc^	0.21 ± 0.05 ^c^	0.01

a,b,c: values with the same letter per row are not significantly different (*p* > 0.05).

**Table 5 vetsci-06-00030-t005:** Effects of potassium dichromate on hematological parameters in rabbit doe.

Blood Parameters	Doses of Potassium Dichromate (mg/kg bw)				*p*
	0	10	20	40	
**WBC (× 10^3^/µL)**	12.43 ± 1.61 ^b^	12.83± 1.69 ^b^	14.17 ± 1.25 ^ab^	16.37 ± 2.14 ^a^	0.04
**Lymphocytes (× 10^3^/µL)**	6.40 ± 0.69 ^b^	5.67 ± 0.47 ^b^	7.97 ± 0.91 ^a^	9.50 ± 1.11 ^a^	0.00
**Monocytes (× 10^3^/µL)**	1.97 ± 0.38	1.77 ± 0.31	2.07 ± 0.49	2.27 ± 0.49	0.57
**Granulocytes (× 10^3^/µL)**	4.07 ± 0.21 ^b^	5.40 ± 0.46 ^a^	4.47 ± 0.55 ^ab^	4.93 ± 1.01 ^ab^	0.05
**RBC (× 10^6^/µL)**	5.28 ± 0.39	5.21 ± 0.30	4.99 ± 0.35	4.56 ± 0.61	0.23
**Haematocrit (%)**	12.47 ± 0.65	12.43 ± 0.70	12.33 ± 0.51	11.27 ± 1.17	0.27
**Haemoglobin (g/dL)**	38.20 ± 2.86 ^a^	38.47 ± 3.72 ^a^	37.03 ± 2.91 ^a^	30.17 ± 4.61 ^b^	0.04
**Platelets (× 10^4^/µL)**	31.83 ± 3.67 ^a^	19.47 ± 2.22 ^b^	22.80 ± 3.48 ^b^	23.77 ± 4.14 ^b^	0.01
**Plaquetocrit (%)**	0.32 ± 0.08	0.21 ± 0.07	0.24 ± 0.07	0.25 ± 0.08	0.36

a,b: values with the same letter per row are not significantly different (*p* > 0.05). RBC: red blood cell count, WBC: white blood cell count.

**Table 6 vetsci-06-00030-t006:** Effects of potassium dichromate on oxidative stress markers in rabbit does.

Organs	Oxidative Stress Markers	Dose of Potassium Dichromate (mg/kg bw)				*p*
		0	10	20	40	
**Liver**	**SOD (µM/min/g of tissue protein)**	3.79 ± 0.78 ^a^	1.79 ± 0.36 ^b^	1.58 ± 0.46 ^b^	1.31 ± 0.26 ^b^	0.00
	**CAT (µM/min/g of tissue)**	10.25 ± 1.91	10.04 ± 1.45	9.35 ± 1.39	9.58 ± 1.07	0.73
	**GSH (µM/g of tissue)**	127.37 ± 28.25	114.32 ± 36.32	106.62 ± 32.64	100.44 ± 14.21	0.60
	**MDA (µM/g of tissue)**	0.98 ± 0.24 ^b^	1.29 ± 0.32 ^ab^	1.41 ± 0.28 ^ab^	1.56 ± 0.48 ^a^	0.05
**Kidney**	**SOD (µM/min/g of tissue protein)**	1.94 ± 0.25 ^a^	1.98 ± 0.24 ^a^	1.61 ± 0.65 ^ab^	1.08 ± 0.47 ^b^	0.04
	**CAT (µM/min/g of tissue)**	10.63 ± 1.38 ^a^	9.31 ± 1.00 ^ab^	9.38 ± 1.54 ^ab^	8.51 ± 1.18 ^b^	0.05
	**GSH (µM/g of tissue)**	87.75 ± 17.04	80.24 ± 17.23	72.19 ± 27.41	68.47 ± 16.49	0.51
	**MDA (µM/g of tissue)**	1.61 ± 0.32 ^c^	2.41 ± 0.53 ^ab^	2.01 ± 0.30 ^bc^	2.66 ± 0.24 ^a^	0.01
**Ovary**	**SOD (µM/min/g of tissue protein)**	6.82 ± 0.69 ^a^	4.65 ± 0.93 ^b^	3.31 ± 0.72 ^c^	2.15 ± 0.55 ^d^	0.00
	**CAT (µM/min/g of tissue)**	13.44 ± 1.77 ^a^	10.97 ± 1.51 ^b^	9.73 ± 1.95b ^c^	7.83 ± 0.66 ^c^	0.00
	**GSH (µM/g of tissue)**	111.76 ± 39.10 ^a^	59.94 ± 13.60 ^b^	79.67 ± 24.23 ^ab^	67.70 ± 17.13 ^b^	0.04
	**MDA (µM/g of tissue)**	1.40 ± 0.40 ^b^	2.40 ± 0.65 ^a^	2.09 ± 0.81^ab^	2.35 ± 0.51 ^a^	0.04
**Uterus**	**SOD (µM/min/g of tissue protein)**	5.81 ± 0.42 ^a^	5.20 ± 0.70^a^	3.96 ± 0.75 ^b^	3.79 ± 0.44 ^b^	0.00
	**CAT (µM/min/g of tissue)**	12.06 ± 2.39 ^a^	9.64 ± 0.37^b^	10.70 ± 1.68 ^ab^	10.06 ± 0.52 ^ab^	0.05
	**GSH (µM/g of tissue)**	79.72 ± 14.26	56.93 ± 18.99	67.96 ± 12.94	57.18 ± 11.81	0.14
	**MDA (µM/g of tissue)**	1.24 ± 0.24 ^b^	1.47 ± 0.18 ^b^	1.63 ± 0.25 ^b^	2.31 ± 0.59 ^a^	0.00

a,b: values with the same letter per row are not significantly different (*p* > 0.05), MDA: malondialdehyde, SOD: superoxide dismutase, CAT: catalase, GSH: glutathione.

## Data Availability

The data sets used during the current study are available from the corresponding author upon reasonable request.

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
