# Peer review of "Oxidative Effects of Potassium Dichromate on Biochemical, Hematological Characteristics, and Hormonal Levels in Rabbit Doe (Oryctolagus cuniculus)"

_vetsci, 2019, doi:10.3390/vetsci6010030_

Round 1

Reviewer 1 Report

I have reviewed the paper titled: “Oxidative effects of potassium dichromate on biochemical, hematological and reproductive characteristics in rabbit doe (oryctolagus cuniculus)”.

The manuscript is interesting and well structured, but revisions are needed before proceeding to acceptance. Language revision is necessary to move forward towards publication.

First of all, I would not mention “reproductive characteristics” in the title as the authors do not actually investigate reproductive performances upon potassium dichromate exposure. I would rather use the term “hormones level”. Indeed, they only analyze blood hormones level and organs weights.

Regarding this issue, I would also like to see some discussion regarding the age of the does and their actual reproductive status, despite them having a relatively undefined oestrus cycle. Moreover, breed of the animals is missing, and this is quite important.

In the abstract the authors says: “divided randomly into 4 equal groups of 7 females each according to body weight”, but then in the methods they state that animals were randomized. The part of the sentence regarding the bw in the abstract is quite confusing. Were the animals simply randomized, or were they divided into groups to equally distribute weights?

The first sentence of the introduction needs some references since authors talk about “many studies”.

LINE 43: reference 2 refers to rats but authors talk about both humans and animals. Can they mention some human references too?

TABLE 1: the table needs to be formatted as the 2 columns are not in line.

Line 79-80: this sentence is not clear…authors talk about 4 experimental groups and a control one…looks like there are 5 groups. I would either refer to “3 experimental groups plus 1 control group” or “4 experimental groups including the control one”.

BIOCHEMICAL ANALYSES: more specifics are necessary since the authors only mention Chronolab kit. This is key for reproducibility.

LINE 99: can the authors mention a reference for the liquid displacement method?

STATISTICAL ANALYSES: I would suggest to perform a simple linear regression analyses to see if there is dose dependence in case of statistical differences. Moreover, since the aim is to compare the treatments with the control group, I would have used the Dunnett’s post-hoc rather than the Duncan’s. The latter is just a suggestion.

TABLE 2: remove letter a from the p-value of urea.

TABLES 2 TO 6: all the tables need to be formatted since are quite confused now and authors have to state what the numbers actually are: means or medians? SD or SEM?

Author Response

Author’s Reply to review report

Comments   and suggestions for Authors (Reviewer 1)

Author’s   Notes to Reviewer

1st paragraph: about the title

Modified according to Reviewer’s suggestion. Please,   see highlighted part of title.

2nd paragraph:

Please, see breed in Line 64: 2.1.Animals and lines   300-304 in discussion.

3rd paragraph:

Animals were divided into 4 groups of 7 females,   with comparable average body weight. Please, see highlighted areas line 14-15   in abstract and line 88-90: 2.3. Toxicological assessment.

4th paragraph: references for 1st   sentence in introduction

Please, see highlighted references in that sentence

5th paragraph: Line 43 about references

Please, see highlighted references: line 52

6th paragraph: Table 1

Please, see modified Table 1

7th paragraph: about number of   experimental groups

3 potassium dichromate treated groups and 1 control   group. That is, 4 experimental groups including the control one. Please, see   highlighted area: line 88-90: 2.3. Toxicological assessment.

8th paragraph: Biochemical analyses

Please, see highlighted area of that section

9th paragraph: Line 99 about reference

Please, the section has been removed as suggested by   Reviewer 2, but this is the reference: ACS (American Chemical Society), 2013.

10th   paragraph: Statistical analyses

Thanks

11th paragraph: remove letter a from P   value

Done

Reviewer 2 Report

This MS explores the effects of potassium dichromate on female rabbit doe, using biochemical, hematological and reproductive approaches. Please include select references including more recent papers namely introduction for a more robust statement of the deleterious effects of this compound. Some omissions were noted as listed below.

Other comments:

Line 13- Twenty eight adult does...

Please include methods on Abstract.

Line 32 – How can authors prove that oxidative stress was a result of the reduction process of chromium?

Line 71 - 2.2. Chemicals

Lines 72-75 – Please remove “Most …crystals”.

Line 75 – Please rephrase the sentence: “K2Cr2O7 is prepared…

Line 79 - Twenty eight animals…

Line 97 – Please remove: “with the help of an electronic …burette”.

Line 97 – Please include uterus.

Line 106 – A volume of 2.0 ml…

Line 113 – 115 - Please rephrase the sentences: “The optical density… 5 min.

Line 226 – exposure instead of exposition

Line 264 - Please remove: White blood cell counts are indicators of the …

Line 304 - Please rephrase the sentence: Additionally, the…

Author Response

Comments   and suggestions for Authors (Reviewer 2)

Author’s   Notes to Reviewer

General comment: recent references in introduction   on deleterious effects of potassium dichromate

Please, see highlighted reference in that   introduction

Line 13

Done

Include methods in abstract

Please, see highlighted area of that section: line   18-25

Line 32

Please, see highlighted area: Line 41 (the statement   is modified to conditional).

Line 71

Done: Line 83: 2.2

Line 72-75

Done

Line 75

Done: line 84

Line 79

Done

Line 97

Done

Line 97

Done

Line 106

Done: line 114

Line 113-115

Done: line 120-122

Line 226

Done: line 242

Line 264

Done

Line 304

Done: conclusion greatly modified

Reviewer 3 Report

The authors described a study investing the toxicity induced by the increasing doses of potassium dichromate in rabbit doe. Overall good design for this study. The paper is very well with defined methodology. However, there are some problems had to be detail clarified 

1. We suggested that the figures were needed to be re-designed again with high resolution

2. Extensive editing of English language s necessary. 
3. The change of body weight during the study in Rabbit Doe is necessary to provided.

4. The result of liver, kidney weight and volume analysis showed significant nephrotoxicity. If authors can provide the pathology of kidney, such as IHC stain or H and E stain, the data will be completed.

5. The bone marrow suppression is the major complication of potassium dichromate. The data of bone marrow analysis is necessary.

6. In this study, the authors studied several important issues and highlighted that potassium dichromate induced considerable changes in female reproductive hormones and generate oxidative stress. A figure of detail conclusion may provide readers easily understand the results.

Author Response

Comments   and suggestions for Authors (Reviewer 3)

Author’s   Notes to Reviewer

1

Done

2

Done

3

Done: Please, see line 149-153 and figure 1 in results and   line 245-248 in discussion.

4

Please, it wasn’t done and we propose as perspective in   order to improve this work.

5

Please, it wasn’t done and we propose as perspective in   order to improve this work.

6

Please, see modified conclusion

Round 2

Reviewer 1 Report

The authors have addressed my concerns and requests.

Minor editings are required:

In the abstract edit line 15-16 to: "Twenty eight adult does of 6 months of age were divided into 4 groups (A, B, C, and D; n=7), with comparable average body weight (bw).

Table 3: enlarge the first column so that "live weight..." and "hepatic tissue..." are distributed on just 1 line.

Figures and tables: letters explanation is weird. I would use: Lowercase letters indicate differences betwen groups (p < 0.05).

Table 6: it is not easy for the reader to discriminate the organs. I would suggest to insert a line after every MDA, so that the separation between the 4 markers for each organ is more clear.

Line 340: Remove "Declare conflicts of interest or state" and just leave "The authors declare no conflict of interest".

Reviewer 3 Report

The authors answered the problems from reviewers by point to point. 

There was no more question for authors.